# Budget impact analysis of a home-based colorectal cancer screening programme in Malaysia

Tran Thu Ngan [1], Kogila Ramanathan,[2,3] Muhamad Raziq Bin Mohd Saleh,[3] Desiree Schliemann [1], Nor Saleha Binti Ibrahim Tamin,[4] Tin Tin Su [2,3], Michael Donnelly,[1] Ciaran O'Neill[1]

¹Centre for Public Health, Queen's University Belfast, Belfast, UK
²Global Public Health, Jeffrey Cheah School of Medicine and Health Sciences, Monash University Malaysia, Selangor, Malaysia
³South East Asia Community Observatory, Monash University Malaysia, Selangor, Malaysia
⁴Ministry of Health Malaysia, Putrajaya, Malaysia

**Correspondence to**
Dr Tran Thu Ngan;
n.t.tran@qub.ac.uk

## ABSTRACT

**Objectives** The 2020–2022 research project 'Colorectal Cancer Screening Intervention for Malaysia' (CRC-SIM) evaluated the implementation of a home-based CRC screening pilot in Segamat District. This budget impact analysis (BIA) assessed the expected changes in health expenditure of the Malaysian Ministry of Health budget in the scenario where the pilot programme was implemented nationwide vs current opportunistic screening.

**Design** Budget impact analysis. Assumptions and costs in the opportunistic and novel CRC screening scenarios were derived from a previous evaluation of opportunistic CRC screening in community health clinics across Malaysia and the CRC-SIM research project, respectively.

**Setting** National level (with supplement analysis for district level). The BIA was conducted from the viewpoint of the federal government and estimated the annual financial impact over a period of 5 years.

**Results** The total annual cost of the current practice of opportunistic screening was RM1 584 321 (~I\$1 099 460) of which 80% (RM1 274 690 or ~I\$884 587) was expended on the provision of opportunistic CRC to adults who availed of the service. Regarding the implementation of national CRC screening programme, the net budget impact in the first year was estimated to be RM107 631 959 (~I\$74 692 546) and to reach RM148 485 812 (~I\$103 043 589) in the fifth year based on an assumed increased uptake of 5% annually. The costs were calculated to be sensitive to the probability of adults who were contactable, eligible and agreeable to participating in the programme.

**Conclusions** Results from the BIA provided direct and explicit estimates of the budget changes to when implementing a population-based national CRC screening programme to aid decision making by health services planners and commissioners in Malaysia about whether such programme is affordable within given their budget constraint. The study also illustrates the use and value of the BIA approach in low-income and middle-income countries and resource-constrained settings.

## STRENGTHS AND LIMITATIONS OF THIS STUDY

⇒ The budget impact analysis (BIA) was used to evaluate the 'affordability' of colorectal cancer (CRC) screening programme in Malaysia within given budget constraint.

⇒ Assumptions and cost inputs for modelling the budget impact were based on the actual costs and rates observed in Malaysia.

⇒ The total cost of resources (=unit costs × number of users) for opportunistic screening and the CRC screening programme were compared with calculate the net budget impact.

⇒ The BIA was conducted from the viewpoint of the federal government and only included costs and resource requirements relevant to this particular budget holder.

⇒ The BIA could not and was not intended to provide answers to questions about whether or not the screening programme is good value for money (which can be answered by a cost-effectiveness analysis).

## INTRODUCTION

Colorectal cancer (CRC) has the second highest incidence and mortality rate among all types of cancer in both sexes in Malaysia.[1] The age-standardised incidence rate in 2012–2016 was 14.8 per 100 000 males and 11.1 per 100 000 females, which appears to be stable compared with 2007–2011.[2] In contrast, the proportion of patients with CRC who are diagnosed at a late stage (ie, stage III or IV) is increasing. Report from Ministry of Health Malaysia (MoHM) showed that the proportion of males with late stage CRC increased from 65.9% during 2007–2011 to 72.4% during 2012–2016; and from 65.2% to 73.1% for females.[2] The report did not give an explanation about this increasing trend though.[2] Late-stage diagnosis negatively impacts survival rate; for example, the 5-year survival rates for cases diagnosed at stage I, II, III and IV in 2002–2004 in Kuala Lumpur were 78.6%, 52.9%, 44.3% and 9.3%, respectively.[3] Improved survival can be achieved by early detection through screening and the removal of premalignant polyps.[4] However,

Malaysia currently does not have a population-based national CRC screening programme.

The MoHM adopted the use of immunochemical faecal occult blood test (iFOBT) for opportunistic CRC screening at public health clinics since 2014.[5] MoHM guidelines recommend screening for asymptomatic individuals aged 50–75 years old with average risk of CRC.[6] The uptake (number of patients screened/total eligible population) of this opportunistic screening tends to be very low. The annual average uptake during 2014–2018 was 0.5% while the 5-year cumulative uptake was 2.29% due to low awareness about CRC in general and CRC tests in particular, fear of the result, concern about the cost and absence of a doctor's recommendation.[5 7] Home-based iFOBT has been implemented in many high-income countries (HICs) to improve the accessibility and uptake of CRC screening.[8] In this context, the Southeast Asia Community Observatory at Monash University Malaysia and Queen's University Belfast (Northern Ireland) collaborated to conduct the research project, 'CRC Screening Intervention for Malaysia' (CRC-SIM) in 2020–2022. This project evaluated the implementation of a home-based CRC screening pilot in Segamat District. The uptake of the novel screening programme was 22%. The significantly higher uptake indicates the potential population wide impact if this screening approach (ie, using home-based iFOBT and self-reporting test results) was scaled up. However, in order to aid public health decision making, there is a need to model a scaled-up version of the research-tested screening programme and, more specifically, gather insights about the total costs of programme implementation and how it might impact the MoHM budget. In other words, there is a need for a budget impact analysis (BIA).

BIA was first introduced in 1998 by Mauskopf.[9 10] Since then, BIA is gradually requested as a part of the health technology assessment (HTA) procedure by a few countries around the world such as Australia, Canada, the the USA, England, Ireland, Spain, Belgium, Poland, Israel and Thailand.[11] Regarding BIA for CRC screening, a recent systematic review found six studies conducted in the UK, USA, Belgium and Australia.[12] We found two additional studies published in 2018 and 2019 from Spain and Thailand, respectively.[13 14] Although results from these studies are not comparable as they were specific to each studied country, all studies were conducted to answer the question 'What is the budget impact of implementing a CRC screening/prevention programme compared with current usual care'. It is also the research question that the BIA in this study aims to answer. Specifically, the BIA assessed the expected changes in the health expenditure of MoHM budget as a result of implementing a population-based national CRC screening programme versus current opportunistic screening (or 'usual care'). It assessed the affordability of the screening programme given potential budget constraints.

## METHODS

The conduct of this BIA and presentation of this paper followed the guidelines developed by the International Society for Pharmacoeconomics and Outcomes Research Task Force.[11 15] All costs are presented in local currency—the Malaysian Ringgit (RM)—and International Dollar (I\$). RM was converted to I\$ using purchasing power parity (PPP) conversion factors instead of market exchange rates. The PPP conversion rate of 1.441 was obtained from the IMF World Economic Outlook Database.[16]

### Health service under assessment and its comparator

The specific health service that was the focus of the BIA was a population-based screening programme for CRC using a self-rapid response iFOBT. The comparator was current or 'usual care'—opportunistic screening.

The BIA is predicated on the opportunistic screening programme being replaced by the new population-based screening programme (ie, the two programmes would not be run in conjunction or in other words, the two scenarios in assessment are mutually exclusive). In each scenario, the patient pathway from the point when patients were invited for screening to receipt of a definitive diagnosis was identified and described. The screening procedure ends at the point of a patient receiving their iFOBT result with encouragement to attend hospital for a colonoscopy (if iFOBT is positive). It is important to note that the BIA included costs of screening and diagnosis (eg, colonoscopy, biopsy) but not treatment. The BIA also did not address issues with respect to equity of access and uptake of services in either screening scenarios.

The patient pathways for the 'usual care' practice and the novel CRC screening programme are presented in figures 1 and 2, respectively. In opportunistic screening practice, it is recommended or expected that individuals who are aged 50–75 years will be screened for CRC symptoms when they attend their local health clinic (for any health condition or problem). If they are asymptomatic and have an average risk of having CRC (based on family history), they are offered an iFOBT, followed by a colonoscopy if the iFOBT test was positive. If CRC is detected following a colonoscopy, the result is conveyed to a patient along with an explanation of the treatment plan or referral arrangement.

Details of the home-based screening intervention in CRC-SIM were published elsewhere.[17] Briefly, in the novel CRC screening programme, individuals aged 50–75 years were contacted, checked for eligibility and invited to participate. A home-screening 'pack' was posted to eligible participants, followed by two reminders. The test was performed at home by participants who took a photograph of the completed test and texted it to trained medical professionals who interpreted the photograph. Participants with positive iFOBT were referred for a colonoscopy at the hospital.

There were two main differences between these patient pathways. First, individuals within the target age

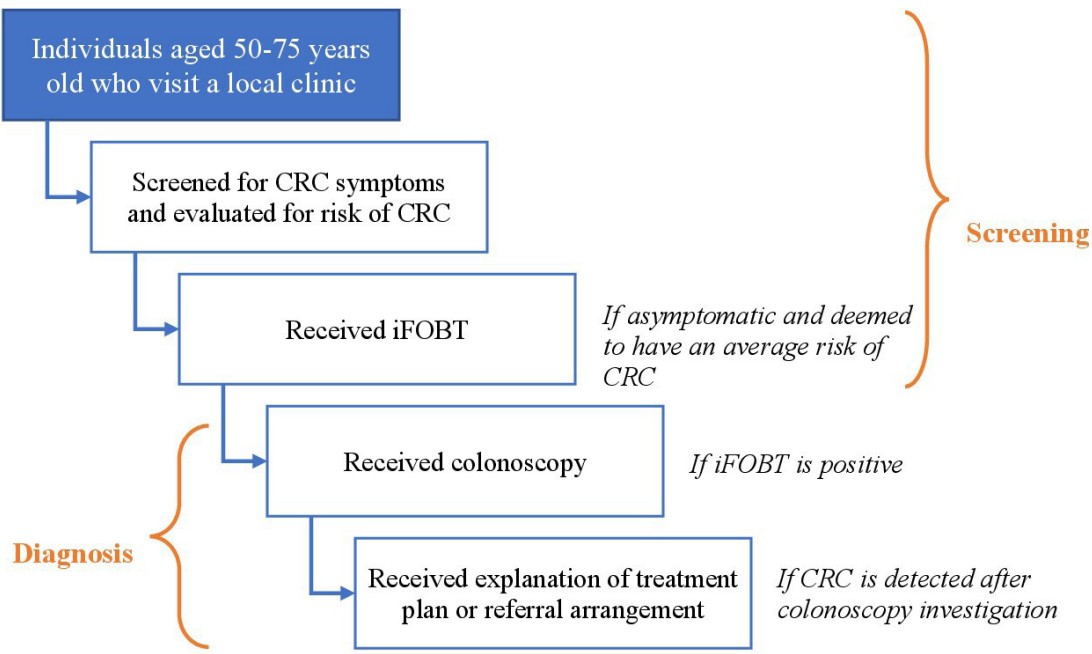

**Figure 1** Patient pathways in 'usual care' practice—opportunistic screening for CRC. CRC, colorectal cancer; iFOBT, immunochemical faecal occult blood test.

group for screening were contacted directly and invited to participate in the novel CRC screening programme while in the situation of 'usual care', CRC screening was offered (if screening guideline recommendations were followed) only when members of the target group visited their clinic for some other health condition or problem. Second, the iFOBT was performed by doctors at health clinics in the 'usual care' pathway while in the novel CRC screening programme, participants self-tested in their home. Home-based testing generated additional stages in the pathways in relation to sending a test, reminding participants, taking a photo of a completed test and sending it to programme officers and vice versa. The remaining stages of each pathway (eg, being screened for eligibility, receiving a colonoscopy and receiving a treatment plan) were the same across the two scenarios.

### Eligible population and input assumptions

The target population for current opportunistic screening in Malaysia is individuals aged 50–75 years, regardless of sex. Due to the nature of home-based screening, the target population for the CRC screening programme was required to meet some additional inclusion criteria as presented in figure 2. The number of individuals who presented and completed each stage was estimated using input assumptions.

Data about the population of Malaysia by age were taken from government reports (ie, Department of Statistics, Malaysia) and from World Population Review.[18 19] The total population was reported to be 32 676 786 in 2021, of which, 19% or 6 228 195 were aged 50–75 years old.[18 19]

### In the 'usual care': opportunistic screening pathway or scenario

All assumptions were derived from a study by Tamin NSI (2020) which was a 5-year evaluation of opportunistic CRC screening (and the use of stool-based tests) in community health clinics across Malaysia.[5] It was assumed that 0.482% of the eligible population would avail of CRC screening when they attended local health clinics for other conditions; and 9.21% of this proportion of tested patients would receive a positive result. Only 55.9% of patients in the study by Tamin availed of a colonoscopy after a positive iFOBT. CRC detection after colonoscopy investigation was 4.04%.

In the novel CRC screening programme, all assumptions were derived from the CRC-SIM research project. It was assumed that 50.51% of the eligible population would be contactable and meet all inclusion criteria to participate in the home-based screening programme; 52.27% of people who were eligible would agree to participate; 41.63% would perform the iFOBT and send a photo of a completed test to the programme officers; 18.01% of people who would be tested would receive a positive result; 41.07% would avail of colonoscopy after a positive iFOBT result and CRC detection after colonoscopy investigation would be 4.35%.

Table 1 summarises details about the input assumptions that were used to estimate the number of individuals at each stage of the respective pathway: the opportunistic screening pathway and the CRC screening programme pathway.

### Cost input and data sources

In the opportunistic screening scenario, the total cost comprised the cost of:

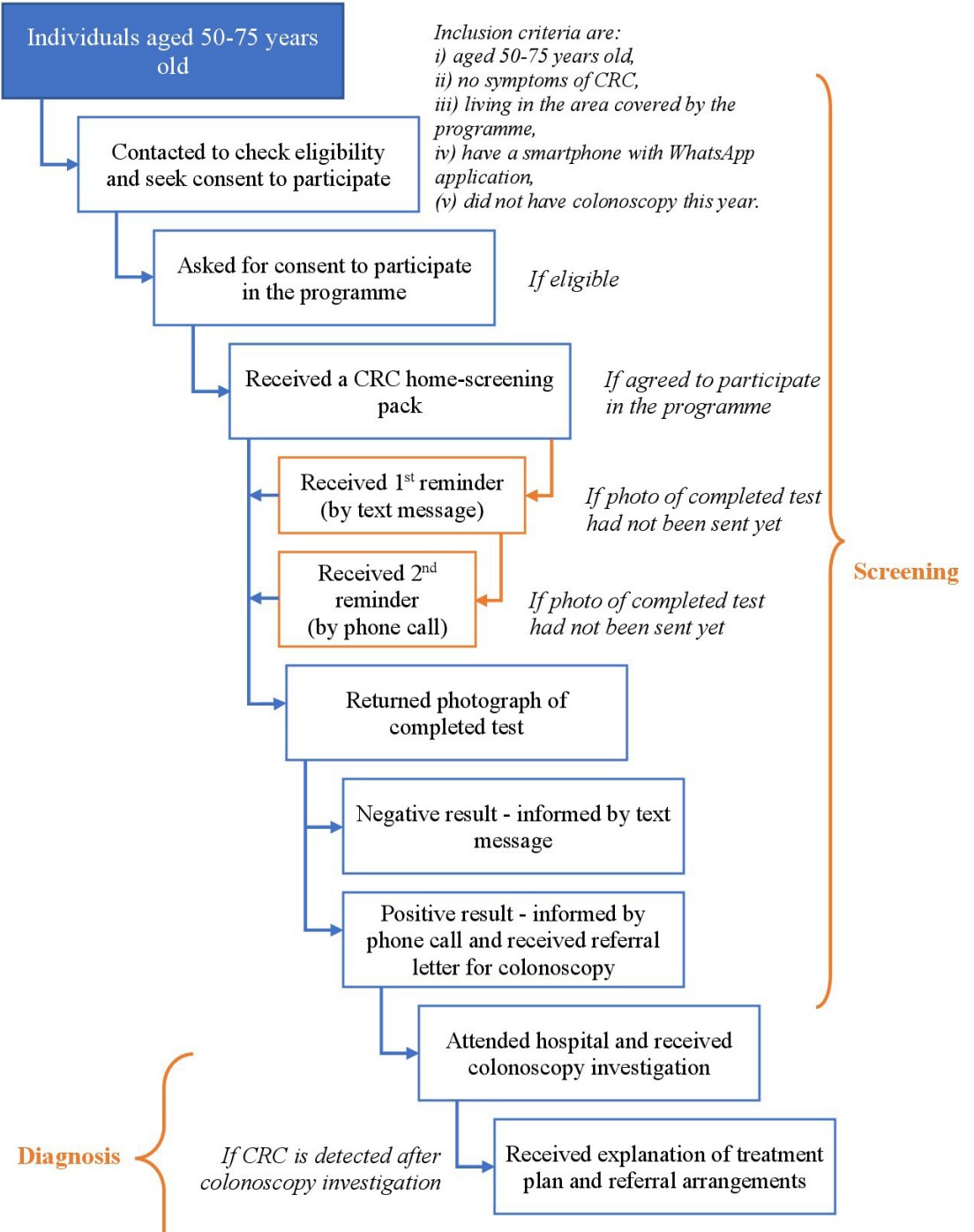

**Figure 2** Patient pathway in population-based CRC screening programme. CRC, colorectal cancer.

1. Performing screening (eg, asking for symptoms, family history and collecting the sample).
2. Processing stool specimens.
3. Interpreting test results.
4. Conveying a definitive diagnosis to patients (include explaining treatment plan or referral arrangements).

In the CRC programme screening scenario, the total cost comprised the costs of:

1. Contacting potential participants.
2. Delivering iFOBT test kits (including cost of the test, postage, print materials and sending video instruction).

3. Sending a reminder to participants (up to two times, by text message and phone call).
4. Interpreting and conveying results to participants.
5. Following up patients with positive iFOBT but did not take colonoscopy in order to encourage them to avail of the colonoscopy.

These costs were calculated by multiplying the time allocated for the completion of each task with the salary cost of the person who undertakes each task plus cost of consumables. Table 2 shows the unit cost for each cost element, related assumptions and data sources.

**Table 1** Input assumptions used to estimate the population at each stage of the patient pathways

| Stage in pathway | Opportunistic screening scenario (current practice) | | Population-based CRC programme screening scenario (proposed practice) | |
|---|---|---|---|---|
| | Assumption* | No of individuals | Assumption† | No of individuals |
| Total population (all ages) | NA | 32 676 786 | NA | 32 676 786 |
| Target population (aged 50–75) | 19.06% | 6 228 195 | 19.06% | 6 228 195 |
| Eligible population (met all inclusion criteria) | 100% | 6 228 195 | 50.51% | 3 146 020 |
| Availed of/agreed to take CRC screening | 0.482% | 30 020 | 52.27% | 1 644 561 |
| Needed first reminder to return the iFOBT result (among those agreed to participate) | NA | NA | 78.71% | 1 294 514 |
| Needed second reminder to return the iFOBT result (among those received first reminder) | NA | NA | 88.10% | 1 140 405 |
| Returned iFOBT result (among those agreed to participate) | 100% | 30 020 | 41.63% | 684 683 |
| Received iFOBT positive result | 9.21% | 2765 | 18.01% | 123 287 |
| Availed of colonoscopy after positive iFOBT | 55.9% | 1546 | 41.07% | 50 636 |
| CRC detection after colonoscopy investigation | 4.04% | 62 | 4.35% | 2202 |

*The assumptions were derived from a study of Tamin NSI (2020) which was a 5-year evaluation of using stool-based test for opportunistic CRC screening in primary health institutions across Malaysia.[5]
†The assumptions were derived from the CRC Screening Intervention for Malaysia research project in Segamat District, conducted by Queen's University Belfast, Monash University and Southeast Asia Community Observatory in 2021.
CRC, colorectal cancer; iFOBT, immunochemical faecal occult blood test; NA, not applicable; No, number.

In the current practice of opportunistic screening, doctors were consulted about the estimated time to perform each stage in the pathway. The monthly salary of a general doctor and a medical laboratory technologist was based on the rate published by the Public Services Commission of Malaysia.[20 21] These rates were RM2947 (~I\$2045) and RM1797 (~I\$1247), respectively.

In the novel CRC screening programme, the time to perform each stage in the pathway, salary of personnel and costs of material resources (eg, rapid kit test, consumables, postage, printing materials) were based on the time and expenditure observed in the CRC-SIM research project. All costs were calculated per screen except the cost of training and the cost of developing communication materials which were one-off costs based on the assumption that communication materials would not change, and no retraining would be needed within 5 years. It was assumed (based on the experience of operating the screening programme during the CRC-SIM project) that one data collector (ie, those employed by the programme to (1) contact potential participants, (2) deliver iFOBT test kits and (3) send a reminder to participants) would be needed for every 400 people in the target population. Training would last 1 day and would be delivered virtually; thus, the cost of training equalled (1-day salary of trainer × number of trainer) + (1-day salary of trainees × number of trainees/data collectors).

### Perspective and time horizon
The BIA was conducted from the viewpoint of the federal government which finances Malaysia's public health system.[22] Only those costs and resource requirements relevant to the budget holder were included in the analysis. For example, the out-of-pocket expenditure incurred by patients were excluded.

The analysis estimated the annual financial impact over a period of 5 years as recommended in the guidelines.[11 23] Costs were not discounted given that the BIA methodology reports the costs for each year in which they occur rather than a net present value.[11]

### Budget impact analyses
#### Computing framework and base-case analysis
The BIA used a cost calculator programmed in Microsoft Excel, following the costing template (The template can be freely downloaded at https://www.nice.org.uk/Media/Default/About/what-we-do/our-programmes/evidence-standards-framework/budget-impact-template.xlsx) produced by the National Institute for Health and Care Excellence in the UK. The template was modified to fit the programme under assessment. The cost calculator approach is recommended by guidelines as it is easy for stakeholders to understand and replicate the results.[11]

First, the number of individuals who completed each stage was estimated (table 1). The resources that were used at each stage of the respective pathways (in opportunistic screening and the novel CRC screening programme) were listed along with their unit costs (ie, cost of each resource per person) (table 2). Unit costs were multiplied by number of users to give the total cost of resources for each scenario. The net budget impact was calculated as the difference in cost between opportunistic screening and the CRC screening

**Table 2** Resources and unit costs

**Currency: Malaysian ringgit (RM)**

| Cost element | Unit cost (Per screen) RM (I$) | Assumptions | Source |
|---|---|---|---|
| **Current practice (opportunistic screening)** | | | |
| Performing screening (asking for symptoms, family history, referral) and taking sample | 5.58 | 20 min × salary RM2947/month | 1 |
| Processing stool specimens | 1.70 | 10 min × salary RM1797/month | 2 |
| Interpreting the test results | 2.79 | 10 min × salary RM2947/month | 1 |
| Conveying a definitive diagnosis to patients (along with explaining treatment plan or referral) | 8.37 | 30 min × salary RM2947/month | 1 |
| **Proposed practice (Population-based CRC screening programme)** | | | |
| Contact eligible individuals—agreed to participate | 0.98 | 7.1 min × (salary RM1440/month+mobile package RM20/month) | 3 |
| Contact eligible individuals—rejected/excluded to participate | 0.47 | 3.4 min × (same as above) | 3 |
| iFOBT rapid test kit | 6.90 | | 3 |
| Print materials (instruction leaflet, explanatory statement) | 1.10 | 90 cents for colour print+20 cent for black and white print | 3 |
| Postage (stamps, etc) | 5.35 | | 3 |
| Sending video through WhatsApp | 0.41 | 3 min × (salary RM1440/month+mobile package RM20/month) | 3 |
| Sending reminder text message | 0.41 | 3 min × (same as above) | 3 |
| Reminder call | 0.28 | 2 min × (same as above) | 3 |
| Interpreting the test kit result | 1.70 | 10 min × salary RM1797/month | 3 |
| Sending text message to inform patient of negative result | 0.45 | 2 min × (salary RM2350/month+mobile package RM20/month) | 3 |
| Calling patient to inform him/her of positive result | 0.67 | 3 min × (same as above) | 3 |
| Preparing and sending referral letter to patient/clinic | 1.12 | 5 min × (same as above) | 3 |
| Follow-up effort | 6.73 | 30 min × (same as above) | 3 |
| Developing communication materials, one-off cost | 6063 | Communication materials do not change in 5 years | 3 |
| Training for data collectors*, one-off cost *Data collectors are those employed by the programme to (1) contact potential participants, (2) deliver iFOBT test kits and (3) send a reminder to participants | 109 703 | + 1 day training (virtual using Zoom) + 1 trainer for maximum 25 trainees + 1 data collector* is needed for every target population of 400 + Cost=1-day-salary of trainer/trainees × number of trainer/trainees + No retraining in 5 years | 3 |
| **Same in both scenarios/practices** | | | |
| Colonoscopy (including polyps removal and/or biopsy if needed) | 200 | | |
| Consumables—stool container, gloves, mask, plastic waste bag and disposal of materials from the test | 10.80 | RM8636.7/800 sets | 3 |

Source: (1) Public Services Commission of Malaysia. Medical Officer Grade UD41. Accessed at https://www.spa.gov.my/spa/laman-utama/gaji-syarat-lantikan-deskripsi-tugas/ijazah-sarjana-phd/pegawai-perubatan-gred-ud41. (2) Public Services Commission of Malaysia. Medical laboratory technologist Grade U29. Accessed at https://www.interactive.jpa.gov.my/ezskim/klasifikasi/perbekalanskim.asp?id_skim=3LU03. (3) CRC Screening Intervention for Malaysia research project in Segamat District, conducted by Queen's University Belfast, Monash University and Southeast Asia Community Observatory in 2021.
CRC, colorectal cancer; iFOBT, Immunochemical faecal occult blood test.

programme. Visual depiction of the cost calculator is shown in online supplemental material, figure S1.

### Uncertainty and scenario analyses

The input assumptions (that were used to estimate the number of individuals at each stage of the respective pathway) and the cost inputs were varied, and then the impact of these changes in relation to the results was analysed to investigate the sensitivity of the budget impact results to variations in individual input. As recommended by Gray et al, the range of variation regarding parameters for which data sources about dispersion were unavailable were ±20% of the base case.[24]

### Patient and public involvement

It was not appropriate or possible to involve patients or the public in the design, or conduct, or reporting, or dissemination plans of our research as this type of study is a secondary analysis of data from a payer perspective (MoHM).

**Table 3** Annual cost of proposed practice (ie, CRC screening programme)

Currency: Malaysian ringgit (RM) and International Dollar (I$)

| Proposed practice | Year 1 RM (I$) | Year 2 RM (I$) | Year 3 RM (I$) | Year 4 RM (I$) | Year 5 RM (I$) |
|---|---|---|---|---|---|
| Contacting adults who are eligible for CRC screening programme (ie, aged 50–75) and screen for eligibility of participating | 2 320 148 (*1 610 096*) | 2 320 148 (*1 610 096*) | 2 320 148 (*1 610 096*) | 2 320 148 (*1 610 096*) | 2 320 148 (*1 610 096*) |
| Providing iFOBT test to adults who agreed to participate in CRC screening programme after being invited | 93 654 886 (*64 992 981*) | 102 612 907 (*71 209 512*) | 111 570 928 (*77 426 043*) | 120 528 949 (*83 642 574*) | 129 486 970 (*89 859 105*) |
| Providing first reminder to participants | 536 929 (*372 609*) | 588 286 (*408 248*) | 639 643 (*443 888*) | 690 999 (*479 527*) | 742 356 (*515 167*) |
| Providing second reminder to participants | 315 339 (*218 833*) | 345 501 (*239 765*) | 375 663 (*260 696*) | 405 825 (*281 627*) | 435 987 (*302 559*) |
| Interpreting returned iFOBT samples | 1 165 129 (*808 556*) | 1 276 572 (*885 893*) | 1 388 016 (*963 231*) | 1 499 460 (*1 040 569*) | 1 610 903 (*1 117 906*) |
| Conveying result through message to participants with iFOBT negative result | 251 990 (*174 872*) | 276 093 (*191 598*) | 300 196 (*208 324*) | 324 298 (*225 050*) | 348 401 (*241 777*) |
| Preparing and sending referral letter and calling participants with iFOBT POSITIVE result | 221 356 (*153 613*) | 242 529 (*168 306*) | 263 701 (*182 999*) | 284 874 (*197 692*) | 306 046 (*212 384*) |
| Following up participants who did not take colonoscopy after positive iFOBT | 489 158 (*339 457*) | 535 945 (*371 926*) | 582 733 (*404 394*) | 629 520 (*436 863*) | 676 308 (*469 332*) |
| Providing colonoscopy (including polyps removal and/or biopsy if needed) to participants with positive iFOBT | 10 127 147 (*7 027 861*) | 11 095 801 (*7 700 070*) | 12 064 455 (*8 372 280*) | 13 033 109 (*9 044 489*) | 14 001 764 (*9 716 700*) |
| Conveying definitive diagnosis to patients (along with explaining treatment plan or referral, etc) after the colonoscopy | 18 432 (*12 791*) | 20 195 (*14 015*) | 21 958 (*15 238*) | 23 721 (*16 461*) | 25 484 (*17 685*) |
| Capital costs (developing communication materials+training for data collectors) | 115 766 (*80 337*) | 115 766 (*80 337*) | 115 766 (*80 337*) | 115 766 (*80 337*) | 115 766 (*80 337*) |
| Total cost of proposed practice | **109 216 279** (*75 792 005*) | **119 429 743** (*82 879 766*) | **129 643 206** (*89 967 527*) | **139 856 670** (*97 055 288*) | **150 070 133** (*104 143 049*) |

Italic numbers inside the brackets indicate values in International Dollar.
CRC, colorectal cancer; iFOBT, immunochemical faecal occult blood test.

## RESULTS

### Base-case analysis

The total annual cost of the current practice of opportunistic screening is RM1 584 321 (~I$1 099 460), of which 80% (RM1 274 690–I$884 587) was for providing opportunistic CRC to adults who availed of the service. Costs of providing colonoscopy (including polyps removal and/or biopsy if needed) after receipt of a positive iFOBT and conveying definitive diagnosis to patients (along with explaining treatment plan or referral, etc) after the outcome of the colonoscopy were RM309 108 (~I$214 509) and RM523 (~I$363), respectively.

The total annual cost over a 5-year period of the proposed practice (ie, CRC screening programme) is shown in table 3. It was assumed that the number of people who would agree to participate in the programme would increase by 5% each year (in consideration of health promotion activities as well as information flows including word of mouth between participants). Therefore, the financial impact would also increase accordingly.

Similar to opportunistic screening, the cost to provide iFOBT to the eligible population who availed of the service accounted for 86% of the total cost of the proposed CRC screening programme. The second most costly component was the provision of colonoscopy (including polyps removal and/or biopsy if needed) to patients with an iFOBT positive result, at 9% of the total cost. The remaining nine cost components such as contacting potential participants, reminding participants to send photograph of iFOBT result, conveying diagnosis to participants and the follow-up effort added only up to 5% of the total cost.

The net budget impact in the first year of implementing CRC screening programme would be RM107 631 959 (~I$74 692 546 which equalled the total cost of future practice minus the total cost of current practice). The impact increases each year as the number of people who agree to participate in the programme increase, reaching RM117 845 422 (~I$81 780 307) in year 2, RM128 058 885 (~I$88 868 067) in year 3, RM138 272 349 (~I$ 95

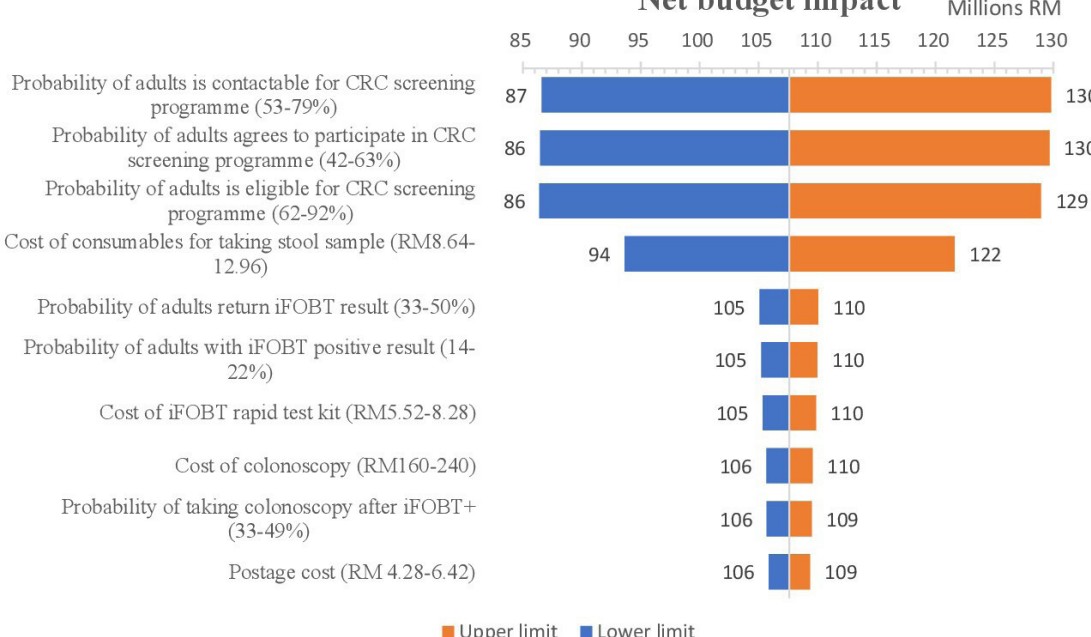

**Figure 3** Results of multiple univariate sensitive analyses showing key factors that exert most influence the net budget impact. CRC, colorectal cancer; iFOBT, immunochemical faecal occult blood test.

955 829) in year 4 and RM148 485 812 (~I$103 043 589) in year 5.

The net budget impact of providing and delivering the CRC screening programme over the 5-year time frame for each state in Malaysia (calculated according to the population size of each state) can be accessed in online supplemental material, table S1. These estimates aid service planning decisions if the novel pilot programme is implemented in one or more of these states before being scaled up into a nationwide programme.

## UNCERTAINTY AND SCENARIO ANALYSES

The tornado diagram in figure 3 shows the change to net budget impact when assumptions and cost inputs were varied. It presents the results of multiple univariate sensitive analyses on key inputs that exert the most influence on the net budget impact (see online supplemental material, table S2 for results of multiple univariate sensitive analysis on all inputs). These inputs include the probability of (1) making successful contact with adults about the CRC screening programme, (2) adults agreeing to participate, (3) adults being eligible to participate in the programme and (4) the cost of consumables that are required to take a stool sample. The first three inputs influence the number of individuals who are present at each stage of the patient pathway.

The net budget impact would increase from RM107 million to RM130 million (~I$74–90 million) if there was a 20% increase in (1) the probability of adults who were contactable (from a contact list of people aged 50–75 years old) or (2) the probability of adults agreeing to participate in the CRC screening programme or (3) the

probability of adults being eligible for the programme (ie, aged 50–75 years old; having no symptoms of CRC, a smartphone and WhatsApp; resident within programme area; and did not have colonoscopy this year). In other words, a 20% increase in each one of these factors would require an additional RM23 million (~I$16 million) to be budgeted for the programme. Likewise, a 20% increase in the cost of the consumables that are required for taking stool samples would mean that the programme would cost RM15 million (~I$10 million) more than the originally calculated total cost.

## DISCUSSION

The result of this analysis provides information to guide public health service planners and commissioners in their decisions about an alternative CRC screening strategy that is, a population-based CRC screening programme using home-based iFOBT compared to current opportunistic screening. It concluded that the net budget impact in the first year of implementing a CRC screening programme of this kind would be RM107 631 959 (~I$74 692 546). The impact would increase by year due to increase in uptake and would reach RM148 485 812 (~I$103 043 589) in the fifth year of implementation. This analytical approach and the results of this analysis are presented as aids to better decision making by MoHs and stakeholders in low-income and middle-income countries (LMICs) about health programme planning and in this particular illustrative case to the MoHM regarding the degree to which the proposed CRC screening programme is affordable.

The total budget that was allocated to the MoHM in 2022 was RM32.4 billion (~I$22.5 million).[25] Spending on prevention and public health services in 2009 was

reported to be RM1.6 billion (~I\$1.1 million).[22] More recent data and information about the size of the budget that is allocated to cancer screening is not available. As such, it is estimated that the net budget impact of implementing a CRC screening programme would account for between 7% and 10% of the total budget for prevention and public health services. This represents a significant proportion of the overall budget allocated for prevention programmes/interventions.

The key factor in the implementation of a population-based screening programme/service or the factor that has biggest impact on the budget is the size of the population who use the service. The degree of accuracy regarding population size estimates is related closely to the cost estimates in the budget. It is important for service planners to keep this point in mind and to take into account an increase in uptake and the impact of such an increase. Therefore, in the case of the CRC programme presented here, we assumed a 5% increase annually in uptake and calculated the net budget impact. The net budget impact can be recalculated according to the actual change in uptake after the programme is implemented.

BIA is an economic assessment that is used to estimate the changes in expenditure of a specific budget holder if a new health technology/programme is implemented.[11] As such, BIA complements other health economic evaluation methods such as cost-effectiveness analysis (CEA) to provide a comprehensive economic assessment of a healthcare intervention to decision-makers.[11] A BIA aids decision making by health service planners and commissioners about whether an intervention or programme is affordable within given budget constraints while a CEA informs decisions about whether an intervention is good value for money.[11 26] BIA and its pragmatic approach is an ideal method when a situation calls for an evaluation of 'affordability' which is of central importance in LMICs and, arguably, is the key concern of whoever is in charge of managing a healthcare budget.[27 28]

It could well be that savings in earlier treatment would counterbalance the additional budget impact. Likewise, reduction in travel and time costs of participant while using home-based screening would reduce the total costs of the screening programme from a societal perspective. If we assume that travel distance to a clinic is 10 km (77% of Malaysian live within 5 km of a clinic[29]), travel time is 10 min (travel speed=60 km/hour), opportunistic screening takes 40 min (table 2) and performing iFOBT at home take 10 min, the reduction in travel and time costs will be 40 min. This can be monetised using gross domestic product per capita at RM50 224[30] to which is then added 10 km x RM1 per km (ie, tolls and fuel[31]) = RM14 per participant. Consistent with BIA best practice guidance these have not been included in our estimate of the BIA which focuses on costs to the provider. Further work in this area may though be useful or an HTA given the potential for aspects of societal cost to influence cost-effectiveness and service uptake.

Finally, the conduct of BIA in this paper has some limitations. First, assumptions and cost inputs for the CRC screening programme were based on the costs and rates that were observed in the CRC-SIM research project. Due to unavailability of data about dispersion of the parameters, the used range of variation (±20% of the base case) may overestimate the uncertainty and suggests that the next step for further research is a CEA where parameter uncertainty is investigated with actual data. The project was conducted in only one district (Segamat); and the distribution of three main ethnic groups (ie, Malay, Chinese, Indian) in the project differed from the proportions that have been reported nationwide (72%:24%:3% vs 62%:21%:6%, respectively). Therefore, it is important to be mindful of the possibility that the assumptions and inputs (based on the project) may not be representative for, or read across to, the whole population of Malaysia. Likewise, it is important to bear in mind that our findings do not include the perspective of other payers and may not generalise to other settings. The results are related directly to the context of the Malaysian health system and the epidemiology of CRC in the country though they are illustrative of the positive contribution of the BIA methodology and approach.

## CONCLUSIONS

This study employed a BIA methodology to analyse the costs of a novel CRC screening programme using home-based iFOBT and mHealth vs the current opportunistic screening. The findings estimated the net budget impact of implementing a population-based national CRC screening programme in Malaysia. The modelling estimations are important considerations for health authorities when they are required to decide the affordability of implementing a programme and to aid budgetary planning as well as decision making, generally, about implementation. Our study illustrates the use and value of the BIA approach in LMICs and resource-constrained settings.

**Contributors** TTN: methodology, formal analysis, writing—original draft, writing—review and editing. KR, MRBMS and DS: data curation, investigation, writing—review and editing. NSBIT: resources, writing—review and editing. TTS and MD: conceptualisation, funding acquisition, writing—review and editing, supervision. CO'N: methodology, formal analysis, writing—review and editing, supervision. MD: guarantor.

**Funding** This research was supported by the Medical Research Council (UK) Global Challenges Research Fund (MR/S014349/1) and the National Medical Research Register Malaysia (NMRR ID-21–02045-07G).

**Disclaimer** The funder of the study had no role in study design, data collection, data analysis, data interpretation, or writing of the report.

**Competing interests** None declared.

**Patient and public involvement** Patients and/or the public were not involved in the design, or conduct, or reporting, or dissemination plans of this research.

**Patient consent for publication** Not applicable.

**Provenance and peer review** Not commissioned; externally peer reviewed.

**ORCID iDs**
Tran Thu Ngan http://orcid.org/0000-0003-2771-9878
Desiree Schliemann http://orcid.org/0000-0002-8746-3002
Tin Tin Su http://orcid.org/0000-0003-0387-6406

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
