## [Reviewer comments · BMJ Open]

ARTICLE DETAILS

TITLE (PROVISIONAL)	A budget impact analysis of a home-based colorectal cancer screening programme in Malaysia
AUTHORS	Ngan, Tran; Ramanathan, Kogila; Saleh, Muhamad; Schliemann, Desiree; Ibrahim Tamin, Nor Saleha; Su, Tin; Donnelly, Michael; O'Neill, Ciaran

VERSION 1 – REVIEW

REVIEWER	Shi, Jufang National Cancer Center / National Clinical Research Center for Cancer / Cancer Hospital, Chinese Academy of Medical Sciences and Peking Union Medical College , Department of Cancer Epidemiology
REVIEW RETURNED	02-Nov-2022

GENERAL COMMENTS	Comments to the Author This study employed a BIA methodology to analyze the costs of a novel CRC screening program using home-based iFOBT and mHealth versus the current opportunistic screening. The findings estimated the net budget impact of implementing a population-based national CRC screening program in Malaysia. This is an interesting study, my comments are as follows: Minor revision: 1. It seems that screening in the novel CRC screening program increases the positive rate of FOBT tests, even higher than the opportunistic screening (9 vs 18%). Is the data reasonable? However, the proportion of colonoscopy costs was lower than opportunistic CRC screening. How to explain it?2. Increase uptake of 5% annually is an important assumption in this study. The definition and basis of uptake should be stated.3. The net budget impact of implementing a CRC screening program would account for between 7-10% of the total budget for prevention and public health services. It's an impressive number. If recent information on the size of the budget allocated to cancer screening is not available, older data can also be instructive and recommended to add. It is also recommended to add some budget information for other comparable cancer (e.g., cervical cancer).4. The method of obtaining the upper and lower limits for the different variables in the multiple univariate sensitive analyses should at least be described in the supplementary. It seems that not all variables were $\pm 20\%$ of baselines.
--

REVIEWER	Cressman, Sonya The British Columbia Cancer Agency
REVIEW RETURNED	10-Nov-2022

GENERAL COMMENTS

This study describes a five-year budget impact assessment for the implementation of a home colorectal cancer screening kit versus the comparative scenario “opportunistic screening” in Malaysia. The authors appear to have completed a robust budget impact analysis, aside from some questionable assumptions about the parameters, however it is not presented in a format that is ready for publication yet. Some critical pieces are still missing, which I listed below. Once these foundational components of a manuscript are integrated, the authors may need to rely on peer review to revise their assumptions and run parts of the analysis again. The issue they intend to address—early detection of colorectal cancer through a home-based screening program—is timely and important, especially in Malaysia where survival rates appear to be far worse than other countries. A key strength of the study is its close connection to community based iFOBT studies, and network members in Malaysia and other countries. To begin with the substantial revisions this study deserves, I suggest the following.

Abstract section

- The message conveyed from the results in the abstract section could be clearer—total annual cost of the current practice of opportunistic screening was
 - 16 RM1,584,321 of which 80% (RM1,274,690)
 - o --274,690 RMI does not equate to 80% of 584,321 RMI; could the authors please clarify?
- The conclusions could be modified to indicate the implications to program planning and draw on the results from the BIA>
- Strengths and limitations section summarizes three main points about the BIA method that should be provided as part of the introduction.

Introduction section

- Introduction section paragraph 1—authors could add a sentence about why the stage distribution is shifting to more later stage diagnoses over time, despite stable incidences. Why is the survival so much different for Malaysia versus US—please clarify, with citations, the role that stage distribution has in the Malaysia/US stage-based 5 year survival comparisons.
- A review of the published literature is missing, this may be borrowed from the network that the authors are connected to or other parts of the home CRC screening program which could summarize the main outcomes studies well and perhaps draw some comparisons with the economics literature on colorectal cancer screening.
- I also suggest the authors consider referencing tailored CRC program interventions such as those outlined in this article: Distributional cost-effectiveness analysis of health care programmes--a methodological case study of the UK Bowel Cancer Screening Programme; M. Asaria, S. Griffin, R. Cookson, S. Whyte and P. Tappenden; Health Econ 2015 Vol. 24 Issue 6 Pages 742-54; Accession Number: 24798212 DOI: 10.1002/hec.3058. Although it uses an extension of CEA, the concepts are relevant to the underlying hypothesis that home based iFOBT.

	Methods section (and parts of the abstract)  • Please provide the PPP conversion rate you used. Describe the discount method for conversion to net present value. Note that there is a sentence about not needing to discount future costs because its annually calculated but this isn't correct; costs the authors reference are in 2020 international dollars or RMI. The same costs need to be discounted by 5years if applied to a scenario five years into the future. • Provide more detail on opportunistic screening that give some clues about why it is not effective—are there cultural barriers or is it just not feasible to go into a clinic for the iFOBt for other reasons? Please elaborate. • The 5-year costs of treatment were excluded. This could potentially be a major limitation of the study, Could the authors please elaborate on the rationale? In high income countries some of the CRC drugs can be very costly, so their exclusion from the BIA is surprising. • Please describe the input assumptions more thoroughly regarding the population that was evaluated. • Provide a reference for the World Population Review. • The input parameters regarding uptake and adherence appear to be well considered. Table 1 is a nice, clear summary of what the BIA reflects. • A literature review could also be undertaken to inform the uncertainty analysis—the +/- 20% rule may be a bit too arbitrary. • Why was it not possible to involve patients in this study? Results  • Highlight the main results with clear section headers (general suggestion) Discussion  • In the first paragraph, suggest to summarize the most impactful finding and how this study adds to the published literature. • Revise the sentence on lines 307-309, which is a good point, but as written it makes an implication about value for money, which the authors correctly acknowledge they were not able to address with BIA. • If the uptake of the service is the key cost-driver then why didn't the authors account for the costs of not taking up the service (i.e. higher colorectal cancer treatment costs)? This should be explained in the discussion about the main findings from the study. • Please add a paragraph stating the limitations of the analysis, available data, method and assumptions.
--	---

	 • Conclusions paragraph should draw on the main findings and implication, not discuss the method. Minor points  • Considering the potential hypothesis that iFOBT at home is more accessible, perhaps the cost analysis should account for not needing to take time off work and travel to clinic? • Are doctors time estimates to most robust data available for this parameter, or perhaps there could be a study to reference for this part of the costing? • Convert RMI to international dollars in the abstract.
--	--

VERSION 1 – AUTHOR RESPONSE

RESPONSE TO REVIEWERS' COMMENTS

REVIEWER 1

This study employed a BIA methodology to analyze the costs of a novel CRC screening program using home-based iFOBT and mHealth versus the current opportunistic screening. The findings estimated the net budget impact of implementing a population-based national CRC screening program in Malaysia. This is an interesting study, my comments are as follows:

Minor revision:

1. It seems that screening in the novel CRC screening program increases the positive rate of FOBT tests, even higher than the opportunistic screening (9 vs 18%). Is the data reasonable? However, the proportion of colonoscopy costs was lower than opportunistic CRC screening. How to explain it?

Response: Thank you for your feedback.

- 1) Both the current opportunistic screening and the novel CRC screening programme provide iFOBT to asymptomatic individuals aged 50-75 years. Those who show up in health clinics and present as symptomatic when being assessed for symptoms of CRC are referred to take colonoscopy directly (bypassed the use of iFOBT). These 'symptomatic individuals', therefore, were not included in the equation to calculate positive rate of iFOBT. The asymptomatic population who take iFOBT test in the novel CRC screening programme is estimated as more than 50 times bigger than the population who take opportunistic screening. It is reasonable to assume that the novel screening programme will have a higher positive rate of iFOBT. Note that, CRC detection rates after colonoscopy investigation of these two programmes are similar.
- 2) That the proportion of individuals who availed of colonoscopy after receiving positive iFOBT in novel screening programme is lower than the opportunistic screening (41% vs 56%) may be explained by the higher trust of individuals towards the tests performed by the health clinics compared to home-based tests performed by themselves. In opportunistic screening, doctors break the news of positive results to individuals and persuade them to avail for colonoscopy directly. In novel screening programme, the

programme officers break the news to the individuals and ask them to go to the health clinics for colonoscopy and further consultation. Higher trust in doctors and the immediate counselling may also result in higher avail of colonoscopy in opportunistic screening.

2. Increase uptake of 5% annually is an important assumption in this study. The definition and basis of uptake should be stated.

Response: Thank you for your feedback. We have added the definition of uptake (including the numerator and denominator to calculate the parameter) to the manuscript (Line 51).

We acknowledged that increase in uptake is an important assumption and have stated that 'The net budget impact can be recalculated according to the actual change in uptake after the programme is implemented' in the discussion section.

Scoping review about the implementation of CRC screening interventions in LMICs^{ref1} suggested that various intervention or programme designs may be successful in terms of achieving a FOBT/iFOBT uptake of $\geq 65\%$. The uptake in the pilot was 22% (given it was conducted during Covid pandemic and without accompanied awareness campaign). Therefore, assumption of 5% increase in uptake annually in the first 5 years is reasonable.

^{Ref1}: Schliemann, D., Ramanathan, K., Matovu, N. et al. The implementation of colorectal cancer screening interventions in low-and middle-income countries: a scoping review. BMC Cancer 21, 1125 (2021). <https://doi.org/10.1186/s12885-021-08809-1>

3. The net budget impact of implementing a CRC screening program would account for between 7-10% of the total budget for prevention and public health services. It's an impressive number. If recent information on the size of the budget allocated to cancer screening is not available, older data can also be instructive and recommended to add. It is also recommended to add some budget information for other comparable cancer (e.g., cervical cancer).

Response: Thank you for your feedback. We agree a comparison of the type requested would be informative. However, apart from the budget for prevention and public health services (RM1.6 billion), we could not find details of budget allocation broken down to specific services/groups of diseases. This type of information may not be published and/or not accessible for general public.

4. The method of obtaining the upper and lower limits for the different variables in the multiple univariate sensitive analyses should at least be described in the supplementary. It seems that not all variables were $\pm 20\%$ of baselines.

Response: Thank you for your feedback. We would like to confirm that, with one exception, all variables in the univariate sensitive analyses were $\pm 20\%$ of baselines. The 'Probability of adults needing 2nd reminder' is already 88% at baseline. The upper value used in sensitive analyses could only be set at the maximum 100% (rather than $88 \times 120\% = 106$).

We have included in the supplementary (as suggested) a table shows the lower and upper values used in univariate sensitive analyses for all variables as well as the corresponding budget impact to each value used. Please see table S2 in the Supplementary materials.

REVIEWER 2

This study describes a five-year budget impact assessment for the implementation of a home colorectal cancer screening kit versus the comparative scenario "opportunistic screening" in Malaysia. The authors appear to have completed a robust budget impact analysis, aside from some questionable assumptions about the parameters, however it is not presented in a format that is ready for publication yet. Some critical pieces are still missing, which I listed below. Once these foundational components of a manuscript are integrated, the authors may need to rely on peer review to revise their assumptions and run parts of the analysis again. The issue they intend to address—early detection of colorectal cancer through a home-based screening program—is timely and important, especially in Malaysia where survival rates appear to be far worse than other countries. A key strength of the study is its close connection to community based iFOBT studies, and network members in Malaysia and other countries. To begin with the substantial revisions this study deserves, I suggest the following.

Abstract section

- The message conveyed from the results in the abstract section could be clearer__total annual cost of the current practice of opportunistic screening was
- 16 RM1,584,321 of which 80% (RM1,274,690)
- o --274,690 RMI does not equate to 80% of 584,321 RMI; could the authors please clarify?

Response: Thank you for your question. We think the reviewer may be confused. The total cost is 1,584,321 of which 80% is 1,274,690 (The numbers are not 584,321 and 274,690).

- The conclusions could be modified to indicate the implications to program planning and draw on the results from the BIA

Response: Thank you for your feedback. We have revised the conclusions of the abstract as suggested.

- Strengths and limitations section summarizes three main points about the BIA method that should be provided as part of the introduction.

Response: Thank you for your feedback. Based on the feedback from editor and requirements from the journal for the 'Strengths and limitations section', we have revised the section which now relate specifically to the methods. The novelty, aims, results or expected impact of the study are not summarised here. Strengths and limitations of methods are further discussed in the discussion section.

Introduction section

- Introduction section paragraph 1—authors could add a sentence about why the stage distribution is shifting to more later stage diagnoses over time, despite stable incidences. Why is the survival so much different for Malaysia versus US—please clarify, with citations, the role that stage distribution has in the Malaysia/US stage-based 5 year survival comparisons.

Response: Thank you for your feedback. We reported our observation on proportion of patients diagnosed at stage III or IV during 2007-2016 using the National Cancer Registry Report from Ministry of Health. The report did not comment on the increasing trend. We could not find studies that investigated this trend. Therefore, we could not add why the proportion of late stage is increasing. Instead, we have added the comment that the increasing trend is unclear.

The survival rate is considerably different between Malaysia and US (50% vs 92%) partly due to the higher proportion of late stage at diagnosis in Malaysia (>70% vs ~50% in US) but also probably due to differences in access to treatment between these countries. To illustrate, that 'One of the most important predictors of CRC survival is stage at diagnosis', we have revised the paragraph with Malaysian data of survival rate by stage of cancer at diagnosis. The added information is "the 5-year survival rates for cases diagnosed at stage I, II, III, and IV in 2002-2004 in Kuala Lumpur were 78.6%, 52.9%, 44.3%, and 9.3%, respectively"

- A review of the published literature is missing, this may be borrowed from the network that the authors are connected to or other parts of the home CRC screening program which could summarize the main outcomes studies well and perhaps draw some comparisons with the economics literature on colorectal cancer screening.

Response: Thank you for your feedback. We have conducted and published a scoping review about the implementation of colorectal cancer screening interventions in low-and middle-income countries. However, we did not include findings from this because it does not contribute to the scope of the current study. The research question is whether CRC screening programme is 'affordable' in the context of Malaysia budget rather than the outcomes or cost-effectiveness of a CRC screening programme. This research question is distinct from that of CEA and we do not want to conflate the two which may serve to confuse, distract or mislead readers from the focus of the study.

- I also suggest the authors consider referencing tailored CRC program interventions such as those outlined in this article: Distributional cost-effectiveness analysis of health care programmes—a methodological case study of the UK Bowel Cancer Screening Programme; M. Asaria, S. Griffin, R. Cookson, S. Whyte and P. Tappenden; Health Econ 2015 Vol. 24 Issue 6 Pages 742-54; Accession Number: 24798212 DOI: 10.1002/hec.3058. Although it uses an extension of CEA, the concepts are relevant to the underlying hypothesis that home based iFOBT.

Response: Thank you for your feedback and suggestion. Due to the difference in methods, we only cite studies that specifically used BIA.

Methods section (and parts of the abstract)

- Please provide the PPP conversion rate you used. Describe the discount method for conversion to net present value. Note that there is a sentence about not needing to discount future costs because its annually calculated but this isn't correct; costs the authors reference are in 2020 international dollars or RMI. The same costs need to be discounted by 5years if applied to a scenario five years into the future.

Response: Thank you for your feedback. PPP conversion rate in use was 1.441 - obtained from the IMF World Economic Outlook Database. We have added this detail into the manuscript.

We have followed the guidelines to conduct BIA from ISPOR—The Professional Society for Health Economics and Outcomes Research, NICE - National Institute for Health and Care Excellence (UK) and HIQA - Health Information and Quality Authority. Discounting is needed for CEA as results are presented as net present value. This is not the case for BIA because the budget holder's interest is in what impact is expected at each point of time (or each year in the 5-year-period). Therefore, and in line with recommended best practice for BIA results are not discounted^{ref2}.

Ref2: Sullivan SD, Mauskopf JA, Augustovski F, et al. Budget impact analysis-principles of good practice: report of the ISPOR 2012 Budget Impact Analysis Good Practice II Task Force. Value Health 2014;17(1):5-14. doi: 10.1016/j.jval.2013.08.2291 [published Online First: 2014/01/21]

- Provide more detail on opportunistic screening that give some clues about why it is not effective—are there cultural barriers or is it just not feasible to go into a clinic for the iFOBT for other reasons? Please elaborate.

Response: Thank you for your feedback. Details of the home-based screening intervention in CRC-SIM were published elsewhere^{ref3}. Nevertheless, we added the reasons why the uptake of opportunistic screening was very low (at 0.5%) as following “low awareness about CRC in general and CRC tests in particular, fear of the result, concern about the cost, and absence of a doctor's recommendation” (cited ref #8 in the reference list).

Ref3: Ngan TT, Donnelly M, O'Neill C. Budget impact analysis of a population-based screening programme for colorectal cancer in Malaysia: technical report of a modelling study. July 1 ed. Belfast: Centre for Public Health, Queen's University Belfast, 2022:32

- The 5-year costs of treatment were excluded. This could potentially be a major limitation of the study, Could the authors please elaborate on the rationale? In high income countries some of the CRC drugs can be very costly, so their exclusion from the BIA is surprising.

Response: Thank you for your feedback. We did not include the cost of treatment after consideration of two main points. Firstly, budget for treatment is different from the budget allocated for prevention programmes and as the purpose of BIA is to illustrate the net budget impact to a specific budget holder, in this study case, the one who in charge of budget for prevention and public health services, consideration of treatment costs, strictly lies beyond the scope of our analysis. Secondly, we do not have access to related data (treatment costs and outcomes). We do recommend that “The conduct of

other types of economic evaluations such as a cost-effectiveness analysis would be required to provide a complete and comprehensive set of evidence for decision makers.”

- Please describe the input assumptions more thoroughly regarding the population that was evaluated.

Response: Thank you for your feedback. The target population for screening in Malaysia is individuals aged 50-75 years, regardless of sex. The patient pathways for opportunistic screening and novel CRC screening are presented in Figure 1. Number of individuals (or population to be evaluated) in each step of the two pathways was calculated and presented in Table 1. In the ‘usual care’ – opportunistic screening pathway or scenario - all assumptions were derived from a study by Tamin NSI (2020) which was a 5-year evaluation of opportunistic CRC screening (and the use of stool-based tests) in community health clinics across Malaysia. In the novel CRC screening programme, all assumptions were derived from the CRC-SIM research project. Assumptions related to each step of the two pathways were provided in the text body and Table 1.

If the reviewer requires more specific details on which further information they want to add, we will provide it.

- Provide a reference for the World Population Review.

Response: Thank you for your feedback. Reference for the World Population Review is ref #14 in the reference list.

- The input parameters regarding uptake and adherence appear to be well considered. Table 1 is a nice, clear summary of what the BIA reflects.

Response: Thank you for your feedback.

- A literature review could also be undertaken to inform the uncertainty analysis—the +/- 20% rule may be a bit too arbitrary.

Response: Thank you for your feedback. CRC-SIM project is the first to implement a home-based iFOBT in Malaysia. We followed the textbook guidance^{ref4} for the case where the dispersion of the parameter was not available from data source: set the range of variation to +/-20% of the base case. The choice of +/- 20% is well documented in the literature rather than our randomly chosen number.

^{Ref4:} M.Gray A, Clarke PM, Wolstenholme JL, Wordsworth S. Applied Methods of Cost-effectiveness Analysis in Health Care. Gray AM, Briggs A, editors. New York: Oxford University Press; 2011.

- Why was it not possible to involve patients in this study?

Response: Thank you for your question. We have included an explanation in the ‘Patient and public involvement section’ as following: “It was not appropriate or possible to involve patients or the public in the design, or conduct, or reporting, or dissemination plans of our research as this type of study is a secondary analysis of data from a payer perspective (Ministry of Health Malaysia)”.

Results

- Highlight the main results with clear section headers (general suggestion)

Response: Thank you for your feedback. Results of BIA are total costs of current opportunistic screening, total costs of the novel CRC screening programme, and the net budget impact. Each of this was presented in one paragraph. We think that sub-headings would be superfluous in this case.

Discussion

- In the first paragraph, suggest to summarize the most impactful finding and how this study adds to the published literature.

Response: Thank you for your feedback. We have included in the first paragraph the net budget impact value (which is the main result from BIA) and implications of the analysis: to guide public health service planners and commissioners in their decisions about implementation of population-based CRC screening programme and aid better decision making by MoHs and stakeholders in lower-middle-income countries (LMICs) about health programme planning.

- Revise the sentence on lines 307-309, which is a good point, but as written it makes an implication about value for money, which the authors correctly acknowledge they were not able to address with BIA.

Response: Thank you for your feedback. The sentence has been rewritten. It now read “As such, it is estimated that the net budget impact of implementing a CRC screening programme would account for between 7-10% of the total budget for prevention and public health services. This sum represents a significant proportion of the overall budget allocated for prevention programmes/interventions .”

- If the uptake of the service is the key cost-driver then why didn't the authors account for the costs of not taking up the service (i.e. higher colorectal cancer treatment costs)? This should be explained in the discussion about the main findings from the study.

Response: Thank you for your feedback. Please see our comment above about not including the treatment cost because it is not within the scope of a BIA. We discussed about this limitation in the Discussion section as well (Line 345-351).

- Please add a paragraph stating the limitations of the analysis, available data, method and assumptions.

Response: Thank you for your feedback. This paragraph is at the end of the discussion section (Line 345 – 364).

- Conclusions paragraph should draw on the main findings and implication, not discuss the method.

Response: Thank you for your feedback. Although this is not a methodology paper, it is one of very few that illustrate the use of BIA in LMICs. Therefore, we would like to emphasize the importance and usefulness of this tool with one sentence in the conclusion section.

Minor points

- Considering the potential hypothesis that iFOBT at home is more accessible, perhaps the cost analysis should account for not needing to take time off work and travel to clinic?

Response: Thank you for your feedback. The BIA was conducted from the viewpoint of the federal government which finances Malaysia's public health system. In BIA, only those costs and resource requirements relevant to the budget holder were included in the analysis. Other costs, for example, the out-of-pocket expenditure incurred by patients or indirect cost were intentionally excluded.

- Are doctors time estimates to most robust data available for this parameter, or perhaps there could be a study to reference for this part of the costing?

Response: Thank you for your feedback. The estimates were based on the ‘real-world’ data specific to the situation investigated and in the absence of other estimates represent the best available. In the practice of opportunistic screening, doctors were consulted about the estimated time to perform each stage in the pathway. In the novel CRC screening programme, the time to perform each stage in the pathway were based on the time and expenditure observed in the CRC-SIM research project.

- Convert RMI to international dollars in the abstract.

Response: Thank you for your feedback. We have provided the equivalent in international dollars in the abstract as suggested.

VERSION 2 – REVIEW

REVIEWER	Shi, Jufang National Cancer Center / National Clinical Research Center for Cancer / Cancer Hospital, Chinese Academy of Medical Sciences and Peking Union Medical College , Department of Cancer Epidemiology
REVIEW RETURNED	29-Dec-2022

GENERAL COMMENTS	The four questions I raised last time have been revised or answered by the author. Questions 2 to 4 have been satisfactorily resolved, but I still have some questions for question 1. The authors note that current opportunistic screening results in a subset of the population going directly to colonoscopy. However, the manuscript does not mention information about these people, nor does it specify the cost of these people and whether the cost needs to be borne by the government. If the government bears the cost, please add the corresponding content to the analysis. If not, please explain the above in the main text.
--

REVIEWER	Cressman, Sonya The British Columbia Cancer Agency
REVIEW RETURNED	06-Jan-2023

GENERAL COMMENTS	The authors have made substantial revisions to the manuscript that have improved the clarity and overall communication about an important topic. They have not however, adequately summarized the literature in their introduction to the study (not necessarily do a literature review) and the conclusion in the abstract now needs to be strengthened. It does not convey a specific take-home message for the reader. Their response to my initial review about why a 20% uncertainty interval was selected should be informed by a clinically relevant example from the CRC screening literature or clinical information rather than citing a general method from an economics textbook. On another note the response to the perspective of the analysis was weak. While the scope of a BIA is related to government funding, home-based programs are specifically intended to prevent travel and time costs, addressing this point with a small modification to the methods or discussion sections would be expected in their response. I do not feel the authors have fully completed the revisions, although appreciate that there were a lot of them. Thank you for inviting me to review
--

	this nice study. I would be pleased to continue working with the authors towards completing the required revisions.
--	---

VERSION 2 – AUTHOR RESPONSE

REVIEWER 1

The four questions I raised last time have been revised or answered by the author. Questions 2 to 4 have been satisfactorily resolved, but I still have some questions for question 1. The authors note that current opportunistic screening results in a subset of the population going directly to colonoscopy. However, the manuscript does not mention information about these people, nor does it specify the cost of these people and whether the cost needs to be borne by the government. If the government bears the cost, please add the corresponding content to the analysis. If not, please explain the above in the main text.

Response: Thank you for your feedback. Patients with symptoms of CRC who present at clinics are referred directly to hospital to undertake a colonoscopy (and bypass the use of iFOBT testing). This is under normal diagnosis procedure. The novel CRC screening programme targets only asymptomatic individuals in the population; thus, it does not replace the usual practice in the case of asymptomatic patients. There are no changes in costs for this group. Therefore, they were not included in the BIA.

REVIEWER 2

The authors have made substantial revisions to the manuscript that have improved the clarity and overall communication about an important topic. They have not however, adequately summarized the literature in their introduction to the study (not necessarily do a literature review) and the conclusion in the abstract now needs to be strengthened. It does not convey a specific take-home message for the reader. Their response to my initial review about why a 20% uncertainty interval was selected should be informed by a clinically relevant example from the CRC screening literature or clinical information rather than citing a general method from an economics textbook. On another note the response to the perspective of the analysis was weak. While the scope of a BIA is related to government funding, home-based programs are specifically intended to prevent travel and time costs, addressing this point with a small modification to the methods or discussion sections would be expected in their response. I do not feel the authors have fully completed the revisions, although appreciate that there were a lot of them. Thank you for inviting me to review this nice study. I would be pleased to continue working with the authors towards completing the required revisions.

Response: Thank you for your feedback.

1) Summary of literature in introduction section

We have added to the discussion of the literature on the advantages and disadvantages of BIA, and the development and growth of BIA in research in the introduction section.

We have also noted that there is no BIA study in Malaysia to date. While there are 8 BIAs of colorectal cancer programmes around the world their usefulness for comparative analysis is restricted by differences in the nature of the programmes, the systems and the target groups.

2) Conclusions of the abstract

We have revised the abstract to focus on the implications of the results for program planning and the usefulness of BIA in LMICs. In keeping with recommended practice for the conduct of a BIA, we have not commented in an evaluative manner on the BIA estimates of budget changes because such judgements belong in the domain of the Ministry of Health Malaysia and require consideration and appraisal by the MoH of their actual budget and resource allocation.

3) +/- 20% use in sensitivity analysis

We have 28 parameters in our models; 19 of them are costs related to the specific CRC opportunistic screening and the novel population-based screening hypothesised for Malaysia; 9 parameters are probabilities mostly related to the operation of screening programme (e.g., Probability of adults participate in screening, needing reminder...).

Where information specific to Malaysia is absent, we have used an accepted approach (i.e., a general method from methods textbook) that allows our estimates to be compared with those used in other BIAs. We note the reviewer's objection to this. We have commented in our limitations section that as and when more precise information on the sensitivity and specificity of tests, uptake rates and other parameters used in our analysis becomes available, further analysis to revise the estimates provided here should be undertaken.

The use of +/- 20% in sensitivity analysis, if any, may overestimate the uncertainty and suggest that the next step is a CEA where parameter uncertainty is investigated with actual data.

4) Perspective of the analysis & the inclusion of travel and time costs

BIA takes the perspective of the budget holder as its fundamental purpose is to inform the government about the costs they will need to bear if the CRC screening programme is implemented. Altering the methods is contrary not only to the purpose of doing a BIA (instead of CEA or CBA or other types of economic evaluation) but also the guidance from the government and expert bodies on conducting BIA.

We acknowledge that home-based programme may reduce travel and time costs for participants. The reduction and its impact are worthy of research in its own right but discussion here would serve to distract from the key message which is the budget impact.

We have added this a limitation and suggestion for further research into the discussion section.

VERSION 3 – REVIEW

REVIEWER	Cressman, Sonya The British Columbia Cancer Agency
REVIEW RETURNED	28-Feb-2023

GENERAL COMMENTS	The authors have responded to most revisions or provided a plausible explanation about why they did not respond to revisions. Their responses are adequate, although I would encourage the authors to acknowledge the societal perspective more authentically; that is to add a simple calculation of the time and travel costs for the purpose of journal publication. The article is not intended for the government so the authors can certainly exclude the costs from their reporting for policy reasons, but for publication I do feel that it is appropriate to acknowledge the potential benefit to patients. Stating this as a limitation and then saying that further research is needed doesn't adequately speak to the patient-centred principals of the BMJ.
---

VERSION 3 – AUTHOR RESPONSE

We have added estimations of travel and time costs into this following paragraph within the discussion section

“It could well be that savings in earlier treatment would counterbalance the additional budget impact. Likewise, reduction in travel and time costs of participant while using home-based screening would reduce the total costs of the screening programme from a societal perspective. If we assume that travel distance to a clinic is 10km (77% of Malaysian live within 5km of a clinic [29]), travel time is 10 minutes (travel speed = 60km/hour), opportunistic screening takes 40 minutes (Table 2), and performing iFOBT at home take 10 minutes, the reduction in travel and time costs will be 40 minutes. This can be monetised using Gross Domestic Product per capita at RM50,224 [30] to which is then added 10km x RM1 per km (i.e., tolls & fuel [31]) = RM14 per participant. Consistent with BIA best practice guidance these have not been included in our estimate of the BIA which focuses on costs to the provider. Further work in this area may though be useful or a health technology assessment given the potential for aspects of societal cost to influence cost-effectiveness and service uptake.”